# The Vaginal Microbiota, Bacterial Biofilms and Polymeric Drug-Releasing Vaginal Rings

**DOI:** 10.3390/pharmaceutics13050751

**Published:** 2021-05-19

**Authors:** Louise Carson, Ruth Merkatz, Elena Martinelli, Peter Boyd, Bruce Variano, Teresa Sallent, Robert Karl Malcolm

**Affiliations:** 1School of Pharmacy, Queen’s University Belfast, Belfast BT9 7BL, UK; l.carson@qub.ac.uk (L.C.); p.boyd@qub.ac.uk (P.B.); 2Population Council, One Dag Hammarskjold Plaza, New York, NY 10017, USA; RMerkatz@popcouncil.org (R.M.); emartinelli@popcouncil.org (E.M.); bvariano@popcouncil.org (B.V.); tsallent@popcouncil.org (T.S.)

**Keywords:** controlled release, drug delivery system, silicone elastomer, ethylene vinyl acetate copolymers, thermoplastics, polyurethanes, vaginal microbiome, lactobacillus, *Gardnerella vaginalis*

## Abstract

The diversity and dynamics of the microbial species populating the human vagina are increasingly understood to play a pivotal role in vaginal health. However, our knowledge about the potential interactions between the vaginal microbiota and vaginally administered drug delivery systems is still rather limited. Several drug-releasing vaginal ring products are currently marketed for hormonal contraception and estrogen replacement therapy, and many others are in preclinical and clinical development for these and other clinical indications. As with all implantable polymeric devices, drug-releasing vaginal rings are subject to surface bacterial adherence and biofilm formation, mostly associated with endogenous microorganisms present in the vagina. Despite more than 50 years since the vaginal ring concept was first described, there has been only limited study and reporting around bacterial adherence and biofilm formation on rings. With increasing interest in the vaginal microbiome and vaginal ring technology, this timely review article provides an overview of: (i) the vaginal microbiota, (ii) biofilm formation in the human vagina and its potential role in vaginal dysbiosis, (iii) mechanistic aspects of biofilm formation on polymeric surfaces, (iv) polymeric materials used in the manufacture of vaginal rings, (v) surface morphology characteristics of rings, (vi) biomass accumulation and biofilm formation on vaginal rings, and (vii) regulatory considerations.

## 1. Introduction

The human vagina is a useful and accessible route for local and systemic administration of drugs, and particularly for clinical indications that are directly associated with women’s sexual and reproductive health. Spurred in part by progressive societal changes to attitudes, behaviors and stigmas around the human vagina, the past twenty years has witnessed increased interest among users, clinicians, and the pharmaceutical industry in developing and using vaginal products for therapeutic benefit.

Two different types of polymeric ring device for vaginal use are currently marketed—drug-releasing vaginal rings (VRs) for pharmacotherapy, and ring pessaries for the management of pelvic organ prolapse and urinary stress incontinence. Drug-releasing VRs—the focus of this review article—are torus-shaped devices designed to administer drugs over extended time periods to the human vagina for therapeutic benefit [1,2,3,4]. To date, seven drug-releasing VRs—Estring^®^, Femring^®^, NuvaRing^®^ (and generics EluRyng™, Myring™), Progering^®^, Fertiring^®^, Ornibel^®^ (also known as SyreniRing and Kirkos^®^) and Annovera™ (Table 1)—have reached market, with total estimated annual sales of $1.8 billion, and many others are in preclinical or clinical development [5,6,7,8,9].

Each marketed ring provides either ‘sustained release’ (drug release maintained over an extended period but not at a constant rate) or ‘controlled release’ (drug release maintained over an extended period at constant or near-constant rate) of one or more steroidal drugs for hormonal contraception (either progestin-only or progestin + estrogen combinations), estrogen replacement therapy, or luteal-phase support for assisted reproduction.

In recent years, there has been very significant innovation in drug-releasing rings, mostly driven by efforts to develop (i) antiretroviral-releasing rings for preventing sexually-acquired infection of human immunodeficiency virus (HIV) in women [4,7,10,11,12], (ii) new longer-acting contraceptive ring devices [6,13,14,15], and (iii) new ring designs that extend the range of drug substances that can be effectively administered beyond conventional hydrophobic small molecules (such as steroid molecules) [11,12,16,17,18,19,20].

By comparison, ring pessaries (often simply referred to as ‘vaginal pessaries’ and not to be confused with dissolvable/meltable drug-containing pessaries/suppositories) are non-medicated polymeric devices inserted vaginally to support areas affected by pelvic organ prolapse, a common condition that occurs when the bladder, rectum or uterus drops or bulges into the vagina [21,22,23]. As with many of the drug-releasing VR products, ring pessaries are commonly fabricated from silicone elastomer, although some devices are manufactured using polyvinylchloride and polyethylene. A more detailed overview of the various polymers used in drug-releasing VRs is presented later in this article. Examples of different types of drug-releasing and pessary-type VRs are presented in Figure 1. Although this article will focus primarily on bacterial adherence and biofilm formation on drug-releasing VRs, much of the information and discussion will also apply to ring pessaries.

The vaginal mucus and mucosa in humans are replete with different bacterial species (Table 2) [24,25,26]. These endogenous microorganisms can attach themselves to the relatively hydrophobic surfaces of VRs, and, depending on the duration of use and the microbial environment, can lead to accumulation of biomass and formation of biofilm on the device surface [27,28,29,30,31,32]. For other common indwelling or implantable medical devices, such as urinary catheters, mechanical heart valves, pacemakers, prosthetic joints and contact lenses, biofilm formation poses critical medical risks, including implant-related infection, persistence of infection, and reduced user acceptability [33,34]. By comparison, little is currently known about vaginal biofilm, much less how that biofilm influences VR product characteristics (drug release, mechanical performance, etc.) or impacts upon the vaginal bacterial ecosystem [35,36]. However, VR biofilm is attributed to endogenous microbiota transferring from the vaginal microenvironment onto the device [27,36,37]. This article is intended to provide a review of the existing scientific literature for the human vaginal microbiota, bacterial biofilm in the vagina, and biomass accumulation and biofilm formation on VRs, at a time of increasing interest in both drug-releasing VRs and the vaginal microbiome [4,6,38,39].

## 2. The Vaginal Microbiota: Healthy and Dysbiotic

It has been estimated that the human body hosts ~4 × 10^13^ bacteria and thousands of different bacterial types, with each body site having its own distinctive communities of microorganisms [42]. Rather than being harmful to their host, the human microbiome is understood to play an increasingly critical role in health and disease [43,44]. As a specific compartment of the human microbiome, the vaginal microbiota in healthy women harbors numerous microorganisms, although with significantly lower diversity compared to other body sites, such as the gut [45]. The vaginal microbiome is particularly responsive to hormonal changes and external influences, including puberty, menopause, pregnancy, sexual activity, use and type of contraceptive products, and personal hygiene [24,25,38,46,47,48]. Due to its dynamic and fluctuating nature, it is not possible to define a “normal” composition for the vaginal microbiome that encompasses all women at all stages of life. However, there exists key groups of microbiological species that are found in most healthy vaginal environments, suggesting that these species play an important functional role in a healthy vaginal ecosystem [38]. Therefore, here we will adopt the definition for the “normal” vaginal microbiota as that present in women with no identifiable disease.

What can be considered a disease-causing pathogen is dependent not only on the type of microorganism and its intrinsic virulence, but also its relative dominance. Microorganisms that are normal constituents of the vaginal flora also have the potential to cause symptoms of disease, but require some alteration of the microenvironment in order to do so [41]. *Candida albicans*, group B *Streptococcus*, *Gardnerella vaginalis*, and *Escherichia coli* are common examples of microorganisms isolated from the lower female genital tract [41]. Under normal circumstances, these organisms do not produce symptoms of infection, but have the potential to cause disease depending on the vaginal environment [41]. This contrasts with putative pathogens not ordinarily part of the vaginal microbiota, whose presence is strongly associated with disease. Examples include sexually transmitted infection with *Neisseria gonorrhoeae*, *Listeria monocytogenes*, and *Trichomonas vaginalis.* Specific examples of microorganisms commonly associated with the vaginal microbiome are discussed in further detail below, including their role in the normal and diseased/dysbiotic vaginal environment.

### 2.1. Lactobacillus spp.

In humans, *Lactobacillus* spp. are the dominant microorganism in the healthy human vagina, found in a relative abundance of greater than 70%. Yet, lactobacilli are rarely found in greater than 1% abundance in the vaginal environment of other mammals [49]. Lactobacilli are Gram-positive, rod-shaped, anaerobic bacteria that produce lactic acid via their metabolic action on the various glycogen breakdown products found in the vagina and formed under the influence of estrogen. It is this lactic acid production that results in a healthy vaginal pH of ~4.2 [38,50]. Lactobacilli will adhere to vaginal epithelial cells, out-competing other microorganisms for surface real estate. They also produce soluble compounds, including bacteriocins, that are toxic to other bacterial species [51]. These attributes of lactobacilli contribute to their dominance in the human vagina and protect against infection with pathogenic microorganisms without themselves inducing inflammation [50]. However, studies have shown that not all lactobacilli are equal in this protective capacity. *Lactobacillus crispatus*, *Lactobacillus gasseri*, *Lactobacillus iners*, and *Lactobacillus jensenii* have been reported as the most frequently occurring species in the healthy vagina [52]. *L. crispatus* is associated with a strong protective and anti-inflammatory capabilities, whereas *L. iners* is easily displaced by other species, and is often associated with a dysbiotic environment. The picture is less clear for *L. gasseri* and *L. jensenii*, although these species appear to be less abundant in states of vaginal dysbiosis [53].

### 2.2. Gardnerella vaginalis

*G. vaginalis,* first described as *Haemophilus vaginalis,* a Gram-variable facultatively anaerobic rod, was proposed over half a century ago as the sole etiological agent of bacterial vaginosis (BV), the most common vaginal infection in women of reproductive age [54]. However, its presence in the vaginal microbiome of healthy women has since cast doubt on its virulence and role as a putative pathogen [55]. It has been detected at rates of 14% to 69% in asymptomatic women [56] and has been isolated as the dominant vaginal microorganism of almost all women with BV [57,58,59]. BV is associated with malodorous vaginal discharge, increased vaginal pH, and the presence of “clue cells” (vaginal epithelial cells with a heavy coating of bacteria, that can be observed microscopically in vaginal fluid). These clue cells are explained by the ability of *G. vaginalis* to form biofilms on the vaginal epithelium, providing convincing evidence for the role of the species in this condition [60]. However, its colonization in asymptomatic women, combined with phenotypic variability and limited taxonomic refinement, results in a somewhat complicated and incomplete understanding of the precise role of *G. vaginalis* within the vaginal microbiome.

*G. vaginalis* has been well known to display genotypic and phenotypic diversity with differing virulence potential, and at least four ‘clades’ (or subgroups) within the species are differentially associated with different clinical outcomes [61,62]. In recent years, biotyping and molecular methods have been applied to categorize these subgroups. In 2019, Vaneechoutte et al. formally proposed three new and distinct species based on whole-genome sequencing and biochemical analysis: *Gardnerella piotii*, *Gardnerella swidsinskii*, and *Gardnerella leopoldii* [60].

BV is also associated with increased risk of HIV infection and transmission [63,64,65,66]. Several factors are likely at play here, including decreased levels of hydrogen peroxide-producing lactobacilli, production of mucin-degrading enzymes, increased influx of HIV target cells, elevated levels of proinflammatory cytokines, elevated vaginal pH, and increased expression of HIV in the lower genital tract. Novel drug-releasing ring formulations to treat or prevent recurrence of BV, including multipurpose devices that simultaneously administer antiretrovirals, have been reported [67,68].

### 2.3. Atopobium vaginae and Prevotella spp.

Another species strongly associated with BV is the strict anaerobe *Atopobium vaginae*, which is resistant to metronidazole and may explain why some women suffer from recurrent BV after treatment with this antibiotic. Studies have reported that *A. vaginae* is present in up to 86% of BV samples [69]. The anaerobic species *Prevotella* spp. is also negatively associated with vaginal health [70]; it has been suggested that colonization with *Prevotella*—the most heritable vaginal bacteria—is strongly associated with host genetics [71]. Women with abundant *Prevotella* in their vagina have higher levels of pro-inflammatory cytokines and increased activation of Toll-like receptors leading to downstream signaling for immune surveillance [71]. Interestingly, there is a strong association between obesity and greater abundance of both gut and vaginal *Prevotella* compared to individuals with BMIs in the healthy range [71,72].

### 2.4. Candida albicans

*Candida albicans* is a polymorphic fungus and a member of the normal human microbiome, residing for the most part harmlessly in the oropharynx, gastrointestinal tract, on the skin, and in the vagina of 20–30% of healthy women [73]. The yeast form (blastoconidia) is typically associated with asymptomatic colonization and transmission, while the hyphal (mycelial) form contributes to adherence and mucosal invasion seen in symptomatic disease [74]. During the switching from a commensal to a vaginal pathogen, *Candida* spp. will also produce a range of extracellular enzymes (including proteases, phospholipases, and hemolysins) that are implicated in adherence to and invasion of vaginal epithelial cells [73]. Another important virulence factor is the ability of *Candida* spp. to form biofilms that attach irreversibly to both biotic and abiotic surfaces; this trait is highly dependent on yeast to hyphal morphogenesis [73,75]. Vulvovaginal candidiasis (VVC) is defined as symptoms of inflammation caused by an overgrowth of *Candida* spp., particularly *C. albicans,* without other infectious etiologies [69]. It is estimated that approximately 75% of all women suffer from VVC at least once in their lifetime [76] and it is a common side effect of treatment with broad spectrum antibiotics, with the eradication of commensal bacteria allowing *C. albicans* to dominate the vaginal microbiota [77].

## 3. Biofilm Formation in the Human Vagina and Its Role in Vaginal Dysbiosis

The observation that bacteria form sessile communities on surfaces was first described in the work of Henrici and Zobell in the 1930s [78,79,80]. However, it was not until the late 1970s with the work of Costerton and colleagues that it was recognized and accepted that biofilms represent the predominant mode of bacterial growth in nature, and indeed in infectious disease [81,82]. A biofilm is defined as a microbially derived community constituted by cells attached to a substratum, interface or to each other, embedded in a matrix of extracellular polymeric substances that they have produced, and that exhibit an altered phenotype with respect to growth rate and gene transcription [83]. The formation and development of a biofilm are affected by multiple factors, including the bacterial strain, the properties of the surface and environmental parameters such as pH, nutrient concentration and temperature [84]. Biofilm formation occurs in five stages independent of whether the attachment surface is of a biotic or inanimate nature: (i) initial attachment; (ii) irreversible attachment; (iii) early development of biofilm architecture (microcolony formation); (iv) maturation; and (v) dispersion (Figure 2), as described in detail in Section 4.

Microorganisms included in biofilms behave differently from free planktonic microorganisms. Bacteria in biofilms show increased tolerance to conventional treatments with antibiotics and can more easily evade the immune system of the host. Biofilm formation by probiotic bacteria, such as *Lactobacillus* spp., is considered a beneficial property because it could promote colonization and longer permanence in the mucosa of the host, avoiding colonization by pathogenic bacteria [85]. It has been demonstrated that the extracellular polymeric substances (EPS) produced by some biofilm forming lactobacillus strains is able to inhibit the formation of biofilms by certain pathogenic bacteria [85,86,87]. Indeed, adhesion and biofilm formation are strain properties that reportedly contribute to the permanence of lactobacilli in the human vagina [88]. However, biofilm formation has been studied more in depth in the context of BV. Indeed, the formation of a polymicrobial biofilm is thought to play a key role in the etiology of BV, regardless of the proposed etiological model [89]. *G. vaginalis* is considered the initial colonizer [35], playing a central role in the early adhesion stage and providing a scaffold for other microorganisms in the mature biofilm. BV biofilm contains consolidate core organisms highly specialized for propagation, although it is unclear which are individual symbionts or accidental beneficiaries and which microorganisms belong to the essential core of biofilm [90]. Bacterial adhesion to vaginal epithelial cells may be mediated by interactions between cell appendages (pili, fimbriae or flagella), carbohydrates and cell surface adhesins [91,92,93]. *G. vaginalis* harbors genes encoding type I, II and IV fimbriae/pili, as well as a biofilm-associated protein (BAP) family gene (bapL) [94,95]. *L. iners* and *Peptoniphilus* spp. are thought to assist *G. vaginalis* during the initial adhesion process [96,97]. Indeed, Castro et al. showed that *L. iners* enhanced the adherence of *G. vaginalis* to epithelial cells rather than inhibiting the bacteria [96].

After adhesion and before biofilm maturation, *G. vaginalis* develops microcolonies [99,100] and induces different symbiotic relationships. *A. vaginae*, *F. nucleatum* and *Mobiluncus* spp. may coaggregate with *G. vaginalis* as secondary/tertiary colonizing species, though the mechanism of coaggregation is not yet known [101]. Moreover, uropathogenic bacteria like *E. coli* and *E. faecalis* can co-aggregate with *G. vaginalis,* enhancing its growth [102]. The possible candidate bridge species between early and late colonizers may be *F. nucleatum* [101], which is well known as the bridge species between early and late colonizers in oral biofilms [103,104], but can also reside in the vagina (in both BV and non-BV cases) [105]. Alternatively, numerous other candidate bacteria may coaggregate as late colonizers as these bacteria have shown a synergistic interaction in a dual-species biofilm model: *Actinomyces neuii*, *Bacillus firmus*, *Brevibacterium ravenspurgense*, *Corynebacterium* spp., *E faecalis*, *E. coli*, *Nosocomiicoccus ampullae*, *Prevotella bivia*, *Propionibacterium acnes*, *Staphylococcus* spp., *Streptococcus* spp. and *G. vaginalis* [106]. After successful microcolony formation/coaggregation, *G. vaginalis* may release extracellular DNA (eDNA) that stimulates the EPS matrix production [100]. eDNA is thought to originate from lysed cells and provides structural integrity and stability to several biofilms [106]. Release of eDNA is maximal during the early exponential growth phase of *G. vaginalis* indicating its active role in biofilm formation [100]. In addition to eDNA, the pathogenic strains of *G. vaginalis* (317 and 594) may encode glycosyltransferases (GT) (family I, II and IV) that seem to be involved in EPS production [95]. The fully matured in vivo BV biofilm appears like ‘brickwork’, a highly organized structure without spaces between bacterial cells [107].

The exact dispersal mechanism of polymicrobial BV biofilm is not known. However, it can be hypothesized that both active and passive dispersion occur during BV. Active dispersion may occur when the vaginal environment becomes favorable during menstruation as the vaginal pH is increased by menstrual blood (pH 7.32) [89]. In contrast, passive dispersion (erosion and sloughing) may be the result of biofilm exposure to the shear forces induced by sialidase, glycosulfatase, glycosidase, proteinase, collagenase and fibrinolysins or the menstrual flow [108,109].

## 4. Mechanistic Aspects of Biofilm Formation on Polymeric Surfaces

The low cost, ubiquity and adaptability of polymeric materials have led to their use in wide range of medical, drug delivery and health care devices. However, they are also easily colonized by biofilms. Where the polymer is an indwelling medical device, the result could be a persistent and difficult to eradicate microbial infection [110]. Infection of a polymeric medical device is likely to occur by inoculation with bacteria from the patient’s own microbiota during placement/implantation. Microbial adherence to these foreign bodies will depend on the surface characteristics of the microbial cell and the nature of the polymer surface [111].

Biofilm formulation proceeds in several stages: initial attachment, irreversible attachment, proliferation, maturation, and dispersion (Figure 2) [98]. The processes are common for biofilm attachment on both biotic and inanimate surfaces, in both medical and environmental scenarios. Biofilm in the context of the vaginal microbiota has been discussed in Section 3 above. The following sections review the mechanistic aspects of biofilm formation on polymeric devices.

### 4.1. Bacterial Attachment to Polymeric Surfaces

A prerequisite for biofilm formation on polymeric devices is that the bacteria achieve sufficient proximity to a surface to allow for initial attachment. The mechanisms by which bacteria are transported to a surface include Brownian motion, sedimentation, and convective mass transport [112]. Several forces, both attractive and repulsive, are important. At approximately 10–20 nm distance from a surface, the negatively charged bacterial cell may be repelled by a negatively charged surface. However, this repulsion can be overcome by attractive Van der Waals forces [113]. Surface appendages on the bacterial cell, such as fimbriae, pili and flagella, are important in surface-sensing for many species of bacteria [114]. For example, the staphylococcal surface proteins SSP-1 and SSP-2 have been described as contributing to *Staphylococcus epidermidis* adherence to polystyrene [110] and these surface structures contribute to adherence by providing means for mechanical attachment [98,112] and overcoming the energy barrier to reach the surface [115]. While cell surface appendages have been suggested as one of the most important factors explaining adhesion, the production of the exo-polysaccharide may also contribute through complexing with the surface [116]. Such mechanical attachments allow for the transition from reversible to irreversible attachment, and facilitate short-range forces such as covalent and hydrogen bonding, and hydrophobic interactions [117]. Additionally, it is well established that after placement or implantation of an polymeric medical device, the surface of the polymer can be rapidly modified. This is dependent on the site and is a result of adsorption of host derived proteins, extracellular matrix proteins and coagulation products [118], such as fibronectin, fibrinogen, thrombospondin, laminin, collagen, and von Willebrand factor [119]. Some of these host factors may serve as specific receptors for colonizing bacteria and influence the extent of bacterial surface attachment. Fibronectin, fibrinogen, and laminin are observed to promote adhesion to biomaterials, while albumin and whole serum appear to inhibit bacterial adhesion to polymeric surfaces [120]. Surface roughness of the polymeric material will also have an influence over the propensity of bacteria to attach to the surface. Rough, irregular surfaces generally allow for better bacterial attachment, providing more niches for cells to adhere, and resulting in increased biofilm density [121].

### 4.2. Biofilm Proliferation and Maturation

Once adhered to a surface, cells upregulate the genes involved in matrix production in as little as 12 min, and so the process of biofilm formation begins, culminating with the development of large cellular aggregates encased in a matrix of EPS [114,122]. The exact composition of EPS can vary between species and is still quite poorly characterized in most biofilms. However, it is widely accepted that this sticky matrix serves to protect the bacterial community from external pressures (host immune defenses, antibiotics, disinfectants, etc.), facilitates the transportation of oxygen and nutrients through its numerous water channels, and contributes to the functioning of intercellular signaling (or quorum sensing, QS) molecules that stimulate the growth and development of the biofilm [119]. Autoinducer signals secreted by the biofilm result in the expression of biofilm-specific genes that influence virulence, while eDNA also contributes to intercellular communication, and stabilizes the structure of the biofilm (as discussed in Section 3, eDNA is important for the structure and integrity for *G. vaginalis* biofilms) [119,123]. During maturation of the biofilm, microcolonies of surface-adhered bacterial will develop into microcolonies, approximately 100 µm in size and thickness [119].

### 4.3. Dispersion and Spread

As the biofilm grows and matures, resources become limited and toxic metabolites may accumulate within the biofilm structure. Dispersal is the mechanism by which the biofilm will expand and colonize new surfaces to circumvent stress-inducing conditions. This can occur as single cells breaking free of the biofilm structures, or clumps of cells being sloughed from the biofilm [124]. The process of dispersal has also been referred to as “metastatic seeding” and can result in biofilm infection spreading to other regions of the body, to the bloodstream where serious embolic complications may occur, or other regions of the medical implant in the case of medical device associated infections [125]. Here, the implications for the patient can be a chronic difficult-to-eradicate infection.

### 4.4. Biofilm Formation on Medical Devices

Biofilm formation on implantable medical devices is a significant contributor to the problem of health care-associated infection (HCAIs), and it is widely accepted that ventilator associated pneumonia (VAP), urinary catheter-associated infections (CAUTI), central-line associated septicemia, and joint prothesis-related infections are all attributable to the formation of microbial biofilms on the respective medical device [126]. In these scenarios, patients likely have pre-existing increased susceptibility to infections due to the serious nature of their illness (i.e., patients requiring artificial ventilation or a central line), or the colonized device is placed into an otherwise sterile region of the body (i.e., placement of a Foley catheter within the bladder, or implantation of a joint prothesis). Conversely the human vagina is abundant in its own microbiota, and this natural microbial community contributes a great deal to vaginal health. Biofilms predominant in lactobacilli may not themselves be any cause for concern. Rather, it is suggested that biofilm formation on polymeric VRs would be the result of preferential adherence and biofilm formation on the polymer by those bacterial species associated with vaginal dysbiosis, which is why VRs must be tested for the presence of bacteria on rings compared to vaginal environment [27].

## 5. Polymeric Materials Used to Manufacture Vaginal Rings

To date, only three types of polymeric material have been used in the fabrication of marketed drug-releasing VRs—silicone elastomers (which are crosslinked forms of polydimethylsiloxane), ethylene vinyl acetate copolymers (EVA), and thermoplastic polyurethanes (TPU) (Table 1, Figure 3) [4]. All these materials are non-biodegradable and hydrophobic, such that they neither dissolve nor swell when immersed in aqueous media or inserted vaginally. However, of these materials, only silicone elastomers and EVAs have direct contact with the vaginal mucosa.

Five of the seven marketed VR products (all except NuvaRing^®^ and Ornibel^®^) are manufactured from medical-grade silicone elastomers. Silicone elastomers are soft, flexible rubber-like materials that have a long history of use in topical and implantable products for a wide range of biomedical and pharmaceutical applications [127,128]. NuvaRing^®^ is manufactured using two different grades of thermoplastic ethylene vinyl acetate (EVA) copolymer (9 and 28% vinyl acetate grades), while Ornibel^®^ contains a thermoplastic polyurethane in its drug-loaded core and a 28% vinyl acetate EVA copolymer for the rate-controlling membrane (Table 1) [129,130].

### 5.1. Silicone Elastomers

Two general types of silicone elastomers have been used for fabrication of VRs. Both are based on crosslinking of chemically functionalized dimethylsiloxane polymers, but differ in their cure (crosslinking) chemistries. Addition-cure silicone elastomers are prepared by chemical reaction between a dimethylsiloxane/methylhydrosiloxane copolymer (in which the reactive species is the hydrosilane group –Si–H) and a vinyl-terminated poly(dimethylsiloxane) (in which the reactive species is the vinyl group –Si–CH=CH_2_). Upon mixing of these two materials, and in the presence of a platinum catalyst, a chemically crosslinked elastomer network (Figure 3) is formed during the high temperature injection molding or extrusion process. A different curing chemistry is used to prepare condensation-cure medical-grade silicone elastomers, involving tin-catalyzed reaction between hydroxy-terminated dimethylsiloxane polymers and a tetra-alkoxysilane crosslinking agent [4,131].

Progering^®^, Fertiring^®^ and Estring^®^ contain only addition-cure silicone elastomers; Femring^®^ is prepared using a condensation-cure silicone elastomer; and Annovera^™^ comprises both types, with the two drug cores prepared using a condensation-cure silicone and the rate-controlling membrane using an addition-cure silicone.

The presence of certain functional groups in the drug molecules can lead to cure inhibition and/or drug binding when using addition-cure silicone elastomer systems [132,133,134,135]. Although the condensation-cure crosslinking reaction is compatible with a much wider range of chemical functional groups, the alcohol by-product formed by the curing reaction can be problematic due drug dissolving in the alcohol and being deposited on the device surface during storage, both potentially impacting drug release kinetics. [136,137].

Silicones are generally regarded as one of the most biocompatible materials for mucosal contact or implantation in humans [127,138,139]. However, bacterial adherence and biofilm formation on silicone elastomer devices has been widely reported, as have strategies to further improve its performance [140,141,142,143,144,145,146,147,148,149,150]. For example, silicone elastomer prosthetic voice valves are implanted in the unsterile environment of the esophagus, such that a mixed biofilm of bacteria and yeast forms rapidly, often causing the voice prostheses to fail within 3–6 months [151,152,153]. Various strategies, including surface modification using argon plasma treatment, chemical grafting of perfluoro-alkylsiloxanes, and use of surface-adsorbed biosurfactants, have been shown to be moderately successful in reducing biofilm formation [140,154,155] in experimental studies. However, biofilm-reducing strategies have not previously been implemented in any marketed silicone elastomer implantable device or drug delivery system.

### 5.2. Ethylene Vinyl Acetate Copolymers

For the reservoir-type rings NuvaRing^®^ and Ornibel^®^, the exterior membranes in contact with the vaginal mucosal tissue are EVA copolymers with 9% and 28% vinyl acetate, respectively. A number of experimental drug-releasing rings fabricated from EVA are also reported [8,18,156,157,158,159,160,161,162,163,164,165].

The use of EVA polymers in drug delivery applications has been reviewed recently [166,167]. Biofilm has been widely reported on the surface of EVA medical devices, along with antimicrobial and surface modification strategies for reducing biofilm formation [168,169,170,171,172,173,174]. It is well understood that the surface free energy increases and the equilibrium contact angle decreases with increasing vinyl acetate concentration in EVA copolymers [175,176,177]. XPS data have revealed that the hydrophobic ethylene component is enriched at the surface of higher vinyl acetate samples, while the more polar vinyl acetate component is enriched on the surface when vinyl acetate < 18% [175]. Increasing the ratio of the polar vinyl acetate residues in the copolymer leads to increased hydrophilicity, as measured by decreased water equilibrium contact angles [178,179,180,181], although no published studies describe the relationship between vinyl acetate content and biofilm formation.

### 5.3. Thermoplastic Polyurethanes

TPUs are beginning to emerge as useful materials for VR fabrication, particularly given the very broad range of properties available by manipulating their chemical composition (Figure 3) [20,67,182,183,184,185,186,187,188]. Only one marketed VR product—the contraceptive ring known as Ornibel^®^, SyreniRing, and Kirkos^®^ (Table 1)—contains a TPU material, although the TPU does not contact the vaginal mucosa, since it is only used in the preparation of the inner drug-loaded core. As with silicone elastomers and EVA copolymers, drug release from hydrophobic non-degradable TPUs is governed primarily by drug diffusion in the polymer [189]. Hydrophobic TPUs have been reported for manufacture of experimental vaginal rings, including for delivery of lactic acid [67] and degradable polyurethanes that are more environmentally friendly [190]. Hydrophilic TPU grades are useful for the development of experimental VRs containing water-soluble drug actives; the hydrophilic segments within the TPU polymer slowly absorb water/aqueous medium/vaginal fluid, leading to device swelling and permitting the solubility and release of the incorporated drug compounds [67,183,184,188]. No studies to date have reported the influence of TPUs hydrophilicity on VR biofilm formation.

## 6. Surface Morphology Characteristics of Vaginal Rings

The mechanisms by which surface roughness, surface charge, and relative hydrophobicity/hydrophilicity serve to facilitate biofilm formation were introduced in Section 4, and these characteristics of implantable medical and drug delivery devices have long been considered to play a crucial role in device-associated infection [191,192,193]. In general, hydrophobic and high rugosity surfaces tend to be more susceptible to adherence by microorganisms [194]. Here, we consider five factors that are likely to contribute to the surface characteristics of VRs—(i) the type of polymer (discussed in detail in the previous section), (ii) the method of manufacture, (iii) the particle size of the drug substance(s), (iv) the drug loading, and (v) the extent of drug release. The latter three factors are relevant only for rings in which the drug substance(s) is available at or near the ring surface.

### 6.1. Method of Manufacture: Injection Molding, Extrusion, Casting and 3D Printing

VRs are generally manufactured at elevated temperatures using either injection molding or extrusion processes. For injection molding of silicone elastomer rings, the drug substance(s) is dispersed into the liquid silicone elastomer components and the resulting mixture injected into a heated mold assembly. For commercial and clinical purposes, ring molds are usually fabricated from martensitic stainless-steel grades, such as 420 SS or 440C, and, depending on the machining methods used to form the mold cavities, the VRs will be left with various surface finishes. The roughness of a mold tool is typically measured as the arithmetic average (R_a_) in micrometers of the absolute values of the profile heights over the evaluation length, though the roughness is specified using standards such as SPI-SPE Finish, Diamond compound finish, VDI 3400, ASA B46.1, BS 1134 and ISO 1302. Mold tools often require a high polish equivalent to SPI-SPE A that requires 3000–6000 grit diamond polishing paste. For some VR products, it is considered desirable to have a light texture on the surface to provide grip and here a mold surface equivalent to SPI-SPE D1/2 is specified, usually produced by blasting the surface with abrasive media of 240–320 grit. Due to the highly abrasive nature of silicones, commercial mold tools are often coated with FDA approved nickel/polymer composites that aid part release, resist wear and increase cavity lubricity. While these coatings are typically applied at a thickness of 20–25 µm, they can be polished to the required surface finish after application. For low volume injection molding of prototype VRs in preclinical development, uncoated aluminum mold tools may be used due to their reduced material costs and often these are left with the surface produced by the final machining operation and minimal hand polishing. In all cases, the surface roughness of the mold tool will directly impact the surface morphology of the manufactured rings, which may in turn impact upon biomass accumulation and biofilm formation, as has been reported previously for silicone voice prostheses [195].

Thermoplastic core-type VRs, such as the marketed products NuvaRing^®^ and Ornibel^®^ (Table 1), are prepared by hot melt co-extrusion. One extruder is used to compound the polymer and drug and a second extruder containing drug-free polymer is used to feed a co-extrusion die that simultaneously extrudes the active core coats in a drug-free polymer membrane [164,184,196]. With extrusion, the surface finish of the extrudate is influenced by several factors. Extruded polymers are known to adhere to the die tool upon exit; this can be particularly problematic for drug-loaded polymers where additional processing aids such as non-stick fluoropolymer additives are not permitted. Polymers also exhibit a phenomenon known as die swell in which the extrudate swells transverse to the die upon exit. This is caused by polymer chain relaxation and randomization after leaving the untangling effects of uniaxial flow environment inside the extruder. Die swell is increased by reducing the ‘land length’ of the die, velocity of flow, decreasing melt temperature and the molecular properties of the polymer. The combination of die surface adhesion and die swelling means that only very sharp or non-radiused exits will lead to a clean separation of the strand from the die. This means any die edge wear or polymer build up at the die exit can lead to surface micro tears and increased surface roughness. ‘Shark skinning’—a phenomenon that creates a periodic ridge-like surface and is related to stress concentrations of the polymer melt exiting the die [197]—can also significantly affect surface roughness of an extruded ring. The final process in extruded VR manufacture is ring jointing, whereby the two ends of the co-extruded strand must be brought in contact and welded together. This has the potential to create a rougher area if excess polymer or flash is created and not trimmed properly. The parts of the polymer that are held in the welding jig are also likely to reach above their softening point and could then take on the surface morphology of the clamp surface, similar to an injection molding operation.

Thermoplastic 3D printing technologies have also been reported for the manufacture of prototype drug-releasing VRs [187,198]. With 3D printing or additive manufacturing, objects are produced in discrete layers as opposed to a single continuous monolith, such that the surface roughness of 3D printed rings is very different to that of injection molded or extruded devices. For example, use of a droplet deposition method used to fabricate an antiretroviral-releasing VR produced a surface in which the individual droplets could be visibly discerned [187]. To date, there has been no investigations into the effect that these 3D printed surface morphologies have on biofilm formation in VRs.

A small number of experimental VR devices have been manufactured by a casting method in which a drug-containing liquid is poured into a ring-shaped mold and allowed to solidify either by cooling or evaporation of the solvent [199,200,201]. Although the design and method of manufacturing are too complex and impractical for commercial purposes, these rings can offer the advantages of biodegradability and multifunctionality. For example, Saxena et al. have reported a hydrogel-type contraceptive ring—based on a nanoporous elastomer system comprising poly(1,8-octanediol-co-citrate) and poly(ethylene glycol) dimethyl ether—offering sustained release of a combination of the spermiostatic agents ferrous gluconate, ascorbic acid and mixtures of polyamino and polycarboxylic acids [201,202,203]. Previous iterations of this ring design were based on a dextran-based hydrogel core and a sheath fabricated using different synthetic biodegradable polymers [199,200]. The initial surface characteristics of these rings are likely determined by the mold surface finish and the materials used to fabricate the ring. Given the biodegradable nature of this ring, it is possible that any surface-adhered bacteria would be periodically sloughed off.

### 6.2. Drug Particle Size and Drug Loading

Most drug-releasing VR products are formulated to contain solid crystalline drug substances. In fact, of the marketed rings, only NuvaRing^®^ and Ornibel^®^ contain drugs that are fully solubilized within the polymer(s), attributed to the high temperature manufacturing process and the relatively low drug loadings. As with other types of pharmaceutical solid dosage forms, solid crystalline drugs incorporated into rings are specially prepared to have an average particle diameter in the low micrometer range. These micron-sized drug particles, referred to as *micronized* drug, are produced by a process called *micronization*, which generally involves mechanical milling of the larger crystalline particles commonly produced during drug synthesis [204]. Drug micronization offers two key advantages—quality control over the particle size distribution of the drug powder, and enhanced dissolution rates. For certain thermoplastic VRs, storage of the product under non-optimal temperatures can cause solubilized drug to migrate from the core through the membrane and precipitate as drug crystals on the ring surface [205].

Drug particle size and drug loading are known to impact the surface morphology of matrix-type rings. For example, incorporation of drug into extruded EVA polymers has been shown to influence the surface morphology, particularly at higher drug loadings [206]. EVA matrices containing up to 50% *w*/*w* metoprolol as a model drug resulted in smooth-surfaced extrudates, whereas at 60% metoprolol content shark skin effects were observed [207]. Of the seven marketed VRs, only Progering^®^ and Fertiring^®^ are of matrix-type design, and both containing relatively high concentrations (>10% *w*/*w*) of solid micronized progesterone powder dispersed throughout the silicone elastomer matrix (Table 1). A dapivirine-releasing matrix-type ring for HIV prevention is also scheduled to reach market soon. Additionally, Murphy et al. have recently reported variable surface discoloration (likely due to menstrual blood) of progesterone matrix rings following clinical use [208]. It is not known to what extent the initial presence of high concentrations of drug at the surface of the VR impacts the extent of discoloration.

### 6.3. Extent of Drug Release

With matrix-type VRs, the drug substance(s) are dispersed throughout the entire ring body. If the drug concentration in the ring body is greater than its solubility in the polymer (and setting aside supersaturated drug states, which occur for certain thermoplastic rings [205]), then a fraction of the drug substance(s) will be present in the solid—and usually crystalline—state. This means that solid crystalline drug is also present at the surface of the ring, and it is this drug that first dissolves and releases when the ring is either inserted vaginally or tested for in vitro release. Dissolution of these surface drug particles leaves behind cavities and pores within the polymer ring, the size and number of which will depend upon the initial drug concentration, its particle size, and the extent of drug release. The surface morphology of these rings will therefore change with time as drug release progresses. For example, approximately 4 mg of the initial 25 mg drug loading in the dapivirine ring is released over 28 days [209,210]; however, most of the drug release occurs from the layers at or close to the ring surface, such that micron-sized cavities are produced after the drug particle dissolves and releases. Of course, this phenomenon will not occur with core-type rings since the outer membrane does not normally contain solid crystalline drug (unless it migrates and precipitates there).

## 7. Biomass Accumulation and Biofilm Formation on Vaginal Rings

The various microorganisms present in the vagina can potentially attach and colonize the surface of VRs and lead to biofilm formation. Theoretically, such biofilm could promote further changes in the vaginal microbiome and adversely affect host mucosal defenses and drug release properties [211]. The limited data to date suggest that biofilm formation on VRs does not alter the vaginal microbiome or impact mucosal host defense. Here, we review the reported studies and data available.

### 7.1. NuvaRing^®^

Miller et al. compared the surface of a NuvaRing^®^ device before and after 28-day use in a single volunteer. Despite the very limited scope of this study, scanning electron microscopy (SEM) did show mucus and cellular debris on the surface of used ring samples that had not been rinsed following removal [32]. With rinsing of the used ring, the SEM images showed similar surface appearance to an unused ring, demonstrating that bacteria were removable and had not strongly adhered to the ring surface. Additionally, using SEM, Keller et al. assessed bioerosion and/or build up of biological material on the surfaces of acyclovir-releasing pod-type silicone elastomer rings during safely and pharmacokinetic testing in six women. After seven days continuous use, sporadic clusters of epithelial cells were observed on the ring surfaces, but little or no associated microbial growth. By day 14, large areas of the ring surfaces were covered with a mat of epithelial cells containing islands of polymicrobial biofilm [211].

Following reports of vaginitis among NuvaRing^®^ users (which is not uncommon with VRs) in a multicenter trial conducted from 1997 to 1999 [212], Camacho et al. assessed the in vitro adherence of five different yeast isolates (*C. albicans, C. glabrata, C. parapsilosis, C. tropicalis* and *S. cerevisiae)* from vaginal exudates in an attempt to understand the potential for VRs to influence development or recurrence of vulvovaginal candidiasis (VVC) [30]. Data obtained through SEM, adherence assays, radiolabeled quantification assays, and measurement of cell surface hydrophobicity on NuvaRing^®^ test segments confirmed that the yeasts tested adhered to the ring to different extents. The authors reflected on whether adherence of vaginal yeast to a VR surface could affect development of VVC in some women. However, this study did not include comparative testing of yeasts found on VRs with cultures taken from women prior to or during ring use, which limited interpretation of these data.

Hardy et al. also reported biomass formation on the surface of NuvaRing^®^ devices following 3 week use by Rwandese women, with the density and composition of the biomass correlated with vaginal dysbiosis [27]. Assessment of ring eluates using quantitative polymerase chain reaction (qPCR) showed that *Lactobacillus* genus, *G. vaginalis* and *A. vaginae* were present in 93%, 57%, and 38% of samples, respectively; these species are commonly associated with BV. SEM analysis showed the surface of the rings covered with vaginal epithelial cells and adhered bacteria. The phenotype comprising a loose network of scattered elongated bacteria was associated with vaginal samples scored as Nugent 0–3 (BV-negative), while the phenotype comprising a dense bacterial biofilm with bacilli-matched vaginal samples scored as Nugent 8–10 (indicative of BV). The findings suggest that the status of the vaginal microbiota and the formation or deposition of biomass on VR are inter-related. The authors noted that VRs releasing pregnancy prevention hormones may be important for protection of the vaginal microbiota during ring use; this hypothesis has been promoted by other researchers [213,214,215].

As part of clinical studies testing intermittent and continuous use regimens of NuvaRing^®^ in Rwandan women, Kestelyn et al. reported the following observations following pelvic examination and tests for STIs and vaginal infections: (i) mean Nugent scores decreased with duration of ring use; (ii) prevalence of trichomoniasis was unaffected by ring use; and (iii) incidence of symptomatic vaginal yeasts increased fivefold compared to baseline [216]. Similar studies with NuvaRing^®^ have also previously been reported: Veres et al. showed an improvement of the vaginal microbiota over three cycles of ring use [215]; Davies et al. did not find a change in BV prevalence during continuous ring use over 56 days [217]; and, in contrast to the Veres study, Oddsson et al. observed an increase in Candida infections during 13 weeks of ring use [218]. Recently, Crucitti et al. reported that NuvaRing^®^ use significantly increased concentrations of Lactobacillus species and decreased concentrations of *G. vaginalis* and *A. vaginae* in vaginal secretions, consistent with the measured reduction in mean Nugent scores [36]. The species composition and extent of the biomass accumulated on the rings correlated with the vaginal microbiota and Nugent score, respectively. Using adherence assays and SEM, Chassot et al. demonstrated that the co-existence and ensuing co-aggregation between *C. albicans* and *L. acidophilus* lead to a significant increase in the in vitro adhesion of *C. albicans* and a decrease in adhesion of the lactobacillus to NuvaRing^®^ [219].

As part of a prospective comparative study in asymptomatic women starting contraception, De Seta et al. reported that women who used the combined contraceptive NuvaRing^®^ showed a significant increase in the number of lactobacilli in the vaginal flora and a reduced Nugent score compared to both baseline and oral contraceptive users [213]. This is most likely attributed to the action of the estrogen ethinyl estradiol on vaginal flora [2,214,215].

### 7.2. Ornibel^®^

Although Ornibel^®^ has the same active pharmaceutical ingredients, overall dimensions and appearance as NuvaRing^®^ (Table 1) and provides pharmacokinetic equivalence [130], the polymers used in both the core and sheath are different. With NuvaRing^®^, the rate-controlling outer membrane comprises a 9% vinyl acetate EVA, while that for Ornibel^®^ comprises a 28% vinyl acetate EVA (Table 1). Sailer et al. have evaluated the adhesion of microorganisms in vitro to both ring devices, and reported that adherence of *C. albicans*, and *L. acidophilus* when co-cultured with *C. albicans*, was lower with Ornibel^®^ [28]. The authors attribute the results to differences in the chemical structure of the polymeric membrane and the smoother surface or Ornibel^®^. However, the article cited by the authors to support the claim of a smoother surface for Ornibel^®^ does not provide any such data [130]. The difference in vinyl acetate content in the EVA polymer membranes of these rings is the more likely explanation for the differences in microbial adhesion, particularly since both rings are manufactured by similar co-extrusion methods and vinyl acetate content of EVAs is known to influence surface properties [175,176,177,178,179,180,181,220]. Supporting this hypothesis, Grandi et al. recently demonstrated using SEM that there are no significant differences in measured surface roughness between unused Ornibel^®^ and NuvaRing^®^ devices [221].

### 7.3. Silicone Elastomer Vaginal Rings

Findings of concordance between vaginal and ring culture results have been reported for several studies. Based on a microbiology sub-study embedded in a Phase 3 study for the segesterone acetate/ethinyl estradiol contraceptive vaginal system (Annovera™; a silicone elastomer ring that is used cyclically for a full year/13 cycles; Table 1), Huang et al. described a high level of agreement between organisms cultured from the vagina following one year of use and organisms cultured from the ring surface [37]. Among the 120 participants in this sub-study, H_2_O_2_-positive Lactobacillus dominated the vaginal microbiota with a non-significant prevalence increase from 76.7% at baseline to 82.7% at cycle 6 and 90.2% at cycle 13. Of the 72 participants who had both vaginal and ring cultures at study exit, 5.6% had a positive vaginal culture and 4.2% had a positive ring culture for *Staphylococcus aureus*. Similar findings of concurrence between vaginal and ring culture results were observed in 62 US women who participated in a randomized 12 week placebo ring trial conducted by the microbicides trial network (MTN 005). Compared with a control group of 30 women who did not use a VR, there were no statistically different outcomes between groups for Nugent scores or vaginal culture results [222]. In a one-year study to assess the effects of Annovera™ on the incidence of vaginal infections and changes in the vaginal microbiota, 3.3%, 15% and 0.8% of subjects were clinically diagnosed with bacterial vaginosis, vulvovaginal candidiasis and trichomoniasis, respectively [37]. These incidence rates were not significantly different from those measured at baseline and Nugent scores were largely unchanged. As in previous studies, a strong correlation between vaginal and ring surface microbiota was reported.

Gunawardana et al. observed microbial biofilms on the surface of both tenofovir and placebo silicone elastomer VRs worn for 28 days by female pig-tailed macaques [31]. Large areas of the ring surfaces were covered with monolayers of epithelial cells and two bacterial biofilm phenotypes were found to develop on these monolayers. Similar findings were noted in a follow-on study in women, including an increase in the volume of cells accumulated on rings over time. The authors suggested that an epithelial cell monolayer develops first and subsequently becomes colonized by islands of polymicrobial communities embedded in extracellular material [211]. The relatively low density of clustered microbial communities observed partially explains the lack of an immune response to the rings worn for up to 14 days. In a further clinical study assessing an acyclovir-releasing pod-type ring for potential treatment of recurrent genital HSV, microbial biofilms were readily detected on the ring surface [223]. However, the composition of these sessile communities was like that of the corresponding vaginal microbiome.

Interestingly, this accumulation of surface biofilm has been considered as a potential cumulative measure of VR use adherence, since characterization of the extent of biomass accumulation could give an indication of the total length of time worn [224]. This approach would not require any modification to a ring design for implementation and would be equally applicable to both placebo and active rings in late-stage clinical testing (assuming little or no differences in surface chemistry/morphology with the drug is incorporated). Potential limitations include inter-individual variation in the vaginal microbiota between participants (perhaps due to the presence of pathogens), which could lead to variations in the rate or type of biofilm accumulation that occurs. Additionally, and probably most critically, the removal, manipulation, or washing of the ring by the participant could lead to biofilm removal or a change in the appearance of the biofilm.

### 7.4. Ring Discoloration

It is also worth commenting briefly on the surface discoloration of VRs that can occur during use, since this may be due to or lead to bacterial adherence/biofilm formation. To date, discoloration has mostly been reported for silicone elastomer rings [208,225], which may reflect the greater number of marketed ring products fabricated from silicone compared to thermoplastics (Table 1) and/or a greater propensity for silicone to become discolored. Discoloration of silicone elastomer devices placed in an unsterile environment may be attributed to biofilm formation, since many microorganisms produce pigments. These pigments are often lipophilic and therefore tend to diffuse into the polymeric matrix. It is generally not possible to remove the stains by simple cleaning [226]. The basic requirements for bacterial colonization of polymer surfaces include a non-sterile environment and sufficient water/moisture to support growth of the bacteria. Silicone elastomer (and other polymer) devices stored in air, even moderately humid air, are not prone to bacterial colonization. For example, silicone elastomer samples stored unpackaged under ambient environmental conditions in the UK for 10+ years do not show any visible signs of bacterial colonization. Any discoloration observed, if any, is usually associated with ageing of the polymer, and is usually indicated by a slight yellow appearance.

A very small number of articles within the scientific literature have reported discoloration during use of VRs [208,227,228,229]; in all likelihood, the phenomenon is significantly under-reported. For example, physical analysis of reservoir-type, silicone elastomer VRs containing either progesterone, d-norgestrel or norethindrone showed surface discoloration and brown staining following clinical use [227]. The authors suggested that the discoloration appeared to be subject-dependent rather than related to the duration of in vivo use. All rings used in the clinical study were initially sterilized using ethylene oxide (ring sterilization is neither required nor generally conducted with modern ring devices), suggesting that the discoloration was associated with use in women (e.g., menstruation) and not attributable to microbial contamination during manufacture. Recently, McCoy et al., described surface discoloration of matrix-type silicone elastomer VRs containing either dapivirine only or a combination of dapivirine and levonorgestrel following clinical use which, based on in vitro assessments using simulated vaginal and menstrual fluids, was attributed to exposure to menstrual fluid [225].

## 8. Regulatory Considerations

Drug-releasing VRs are manufactured in clean (but not sterile) environments in which the bioburden levels are controlled to achieve conformity to product requirements. Moreover, just as with other pharmaceutical solid dosage forms, rings are generally not sterilized after manufacture, since the vagina itself is not sterile and the ring product does not contain water or any other substance supportive of microbial growth. As such, drug-releasing VRs are formally classified as ‘non-sterile pharmaceutical dosage forms’ and pharmacopeial monographs for marketed drug-releasing ring products do not include microbiological purity criteria. Overall, the microbial risk assessment associated with drug-releasing VRs is low relative to many other medical devices and pharmaceutical dosage forms.

Although regulatory health authorities sometimes request that drug-releasing VRs undergo clinical evaluation for the presence of biofilms and changes in the vaginal microbiota (e.g., FDA requested the microbiology study that was conducted as a Phase 3 Annovera™ sub-study and clearly have an interest in biofilm formation and its effects), there are currently no guidelines or standardized testing protocols. FDA, NIH and other research organizations have held workshops and open public meetings focused on development of evidence in relation to health care associated infections [230]. Currently, the best approach to providing information about VR biofilms in a clinical study is to answer the following questions: Does the VR destabilize the vaginal microbiome (sustainability of the vaginal microbiome) and promote some sort of dysbiosis such as BV? Does bacterial colonization on a VR surface lead to a vaginal infection? Does biofilm formation on the surface of the ring affect the release of the APIs (biofouling)? One approach for accumulating data on these questions is to identify the vaginal microbiota before device insertion, during insertion and after insertion. The vaginal microbiota can then be characterized on both the vaginal mucosa and the ring device itself. Biofilm testing on a VR is possible upon immediate removal of the ring and subsequent testing, using techniques such as crystal violet binding assay to measure biomass density, scanning electron microscopy and quantitative polymerase chain reaction [27,36,37].

Further studies are recommended to understand clinical implications of biofilm formation on vaginal devices and to clarify expected norms for both vaginal flora and the devices (there are no expected norms presently to guide interpretation of the data). For sexually active women who participated in NuvaRing^®^ clinical trials, vaginal cultures exhibited vaginal flora changes over time. These were expected and showed that ring use was not linked to unhealthy changes in the vaginal microbiota [213,231]. For Annovera™ the same organisms found on the rings were identified in the vaginal cultures and results did not indicate any clinical problems. Similarly, the study conducted with placebo rings revealed concordance between vaginal and ring cultures [222].

Additional questions that arise relate to instructions for caring for rings that are in use, i.e., used continuously or according to specified ring in/ring out intervals. Biofilms develop quickly on ring devices following use in the human body. While instructions for use for most VRs include directions for washing these rings with mild soap and water following periods of use and prior to reinsertion, further research is required to clarify the effect that various washing regimens may have on biofilm formation or the vaginal microbiome upon reinsertion.

## 9. Conclusions

There is very considerable interest in new drug-releasing VR products, primarily driven by ongoing efforts to develop (i) new longer-acting contraceptive VRs, (ii) antiretroviral rings for preventing HIV acquisition, and (iii) multipurpose technology rings offering various combinations of clinical benefits. The various polymers and methods used in the manufacture of VRs means that the products are not sterile. Although ring products are prepared in a clean environment, the devices are not required to be completely free of all microorganisms. However, as with all non-sterile pharmaceutical products, bioburden control is essential, particularly given the potential for the rings to impact—for good or for bad—the healthy vaginal microbiome. Since VRs were first reported in the 1970s, we have accumulated an extensive body of data indicating that rings are highly effective for a range of therapies, are safe to use even over long periods of time, and do not increase the risk of infection. Based on epidemiologic research regarding the benefits of sex hormones on the vaginal microbiome, and results from recent microbiology and biofilm studies with rings containing hormones to protect against pregnancy, vaginal rings may even be protective.

The emerging and rapidly progressing field of microbiome research is paving the way for a better understanding of how the microbiome influences human health and disease. The vaginal microbiome—influenced as it is by a plethora of internal and external factors, including hormonal changes, the menstrual cycle, sexual activity, application of hygiene products, etc.—is rather unique. Despite containing several hundred different types of bacterial species, the healthy human vagina is consistently dominated by a surprisingly small number of Lactobacillus species, presumably having evolved as important to vaginal health and human reproduction. As with the application of any foreign body to the vagina, drug-releasing VRs have the potential to influence the vaginal microbiome, either by introducing exogenous bacteria into the vagina or by providing a surface for adherence of endogenous bacteria and ultimately leading to biofilm formation. Encouragingly, reports are starting to emerge exploring microbial dynamics in this unique ecosystem. However, much more work is needed. For example, we need in situ microbial testing to reinforce current data demonstrating that a healthy vaginal ecosystem is maintained during VR use, and in the case of VRs with hormones may even be protective. Even more intriguing is the possibility of new ring products to actively promote a healthy vaginal ecosystem.

## Figures and Tables

**Figure 1 pharmaceutics-13-00751-f001:**
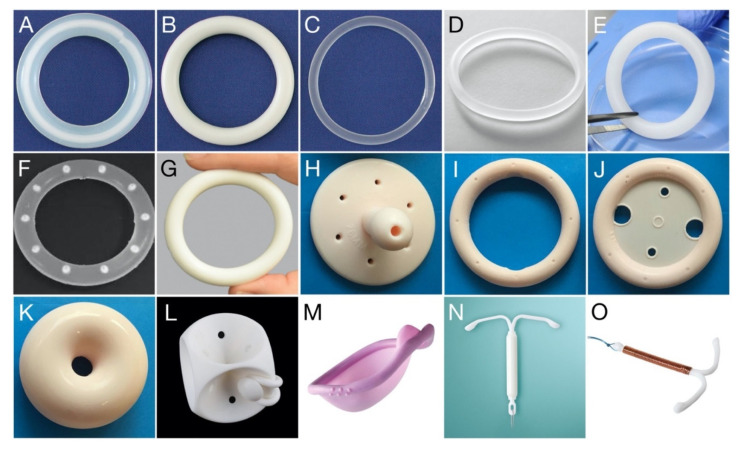
Photo gallery showing various drug-releasing VRs, VR pessaries, and other polymeric devices for vaginal/cervical/uterine administration. (**A**) Estring^®^; (**B**) Femring^®^; (**C**) NuvaRing^®^; (**D**) Ornibel^®^; (**E**) dapivirine-releasing VR, for HIV prevention; (**F**) pod-type VR; (**G**) Annovera^™^; (**H**) Gellhorn pessary; (**I**) Ring pessary without support; (**J**) Ring pessary with support; (**K**) Donut pessary; (**L**) Cube pessary; (**M**) Caya^®^ diaphragm (size: 67 × 75 mm); (**N**) Mirena^®^ intrauterine device; (**O**) Gynefix^®^ intrauterine device. Each vaginal ring devices presented (**A**–**G**) has an overall diameter within the range 54–56 mm; further details are provided in Table 1. Vaginal pessary devices (**H**–**L**) are available in different sizes ranging from 44.5 to 127 mm (Size 0–13).

**Figure 2 pharmaceutics-13-00751-f002:**
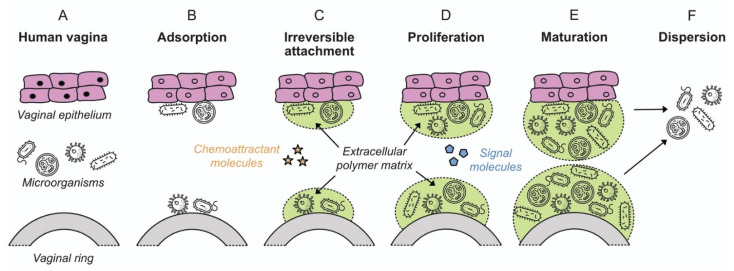
Colonization of vaginal epithelium and a VR device by microorganisms, showing the various stages of biofilm formation (adapted from [98], American Scientist, 2005). Microorganism icons made by Freepik, Wanicon, Darius Dan and Smashicons (www.flaticon.com; accessed 21 February 2020).

**Figure 3 pharmaceutics-13-00751-f003:**
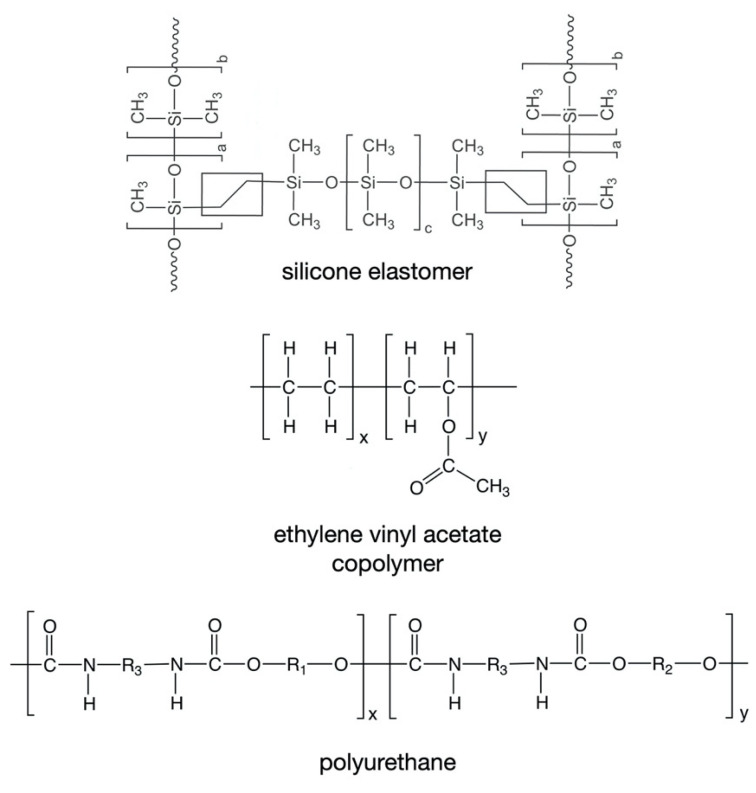
Chemical structures of the polymers commonly used in the fabrication of drug-releasing VRs.

**Table 1 pharmaceutics-13-00751-t001:** Descriptions of marketed vaginal rings.

Vaginal Ring (Company)	Device Type (Duration of Release)	Active Agent(s) (Loading/Release Rate)	Polymer(s)	Indication	Ring Dimensions
Estring^®^ (Pfizer)	reservoir (3 months)	17β-estradiol (2 mg/7.5 μg/day)	silicone elastomer core and sheath (both Q7-4735, Dow)	estrogen replacementtherapy	Ring OD: 55 mm Ring CSD: 9.0 mm Core CSD: 2.0 mm Core length: 145 mm
NuvaRing^®^ (Merck) EluRyng™ (Amneal) Myring™ (Mithra) Generic (TEVA)	reservoir (21 days)	etonogestrel (11.7 mg/120 μg/day) ethinyl estradiol (2.7 mg/15 μg/day)	28% EVA * copolymer core and 9% EVA * sheath	combination contraception	Ring OD: 54 mm Ring CSD: 4.0 mm Membrane thickness: 110 μm
Femring^®^ (Millicent)	reservoir (3 months)	17β-estradiol-3-acetate (12.4, 24.8 mg/50, 100 μg/day)	silicone elastomer core and sheath (both MED-6382, NuSil)	estrogen replacement therapy	Ring OD: 56 mm Ring CSD: 7.6 mm Core CSD: 2.0 mm Core lengths: 8 and 16 mm
Progering^®^ (Population Council/Silesia SA/Grupo Grünenthal Chile)	matrix (3 months)	progesterone (2074 mg/~10 mg/day)	silicone elastomer (MED-4211, NuSil)	post-partum contraception in breastfeeding women	Ring OD: 56 mm Ring CSD: 8.4 mm
Fertiring^®^ (Population Council/Silesia SA/Grupo Grünenthal Chile)	matrix (3 months)	progesterone (1000 mg/~10 mg/day)	silicone elastomer (MED-4211, NuSil)	IVF/hormone supplementation	Ring OD: 56 mm Ring CSD: 8.4 mm
Ornibel^®^ (Exeltis) SyreniRing (Crescent Pharma) Kirkos^®^ (Farmitalia)	reservoir (21 days)	etonogestrel (11.0 mg/120 μg/day) ethinyl estradiol (3.47 mg/15 μg/day)	polyurethane sheath and 28% EVA* copolymer core	combination contraception	Ring OD: 54 mm Ring CSD: 4.0 mm Membrane thickness: 150 μm
Annovera^™^ (Population Council)	reservoir (1 year)	segesterone acetate (103 mg/150 μg/day) ethinyl estradiol (17.4 mg/13 μg/day)	silicone elastomer cores (x2, MED-6603 and MED-6385, NuSil) and sheath (MED-4224, NuSil)	combination contraception	Ring OD: 56 mm Ring CSD: 8.4 mm Core CSD: 3.0 mm Core lengths: 11 and 18 mm

* EVA—ethylene vinyl acetate; ^#^ OD—overall diameter; CSD—cross-sectional diameter.

**Table 2 pharmaceutics-13-00751-t002:** Prevalence of microorganisms reported in asymptomatic vaginal or cervical specimens (adapted from [40], Infectious Diseases: Research and Treatment, 2010 and [41], Clinical Infectious Diseases, 2001). + < 30%; ++ < 60%; +++ > 60%.

**Gram-Positive Rods**		
	Diptheroids	+++
	Lactobacilli	+++
	Gram-positive cocci	
	*Staphylococcus aureus*	+
	*Staphylococcus epidermidis*	++
	*Streptococcus* species	
	α-Hemolytic	+
	Β-Hemolytic	+
	Non-hemolytic	+
	Group D	+
**Gram-Negative Rods**		
	*Escherichia coli*	+
	*Klebsiella* and *Enterobacter* spp.	+
	*Proteus* spp.	+
	*Pseudomonas* spp.	+
**Anaerobic Species**		
	*Bacteroides* spp.	++
	*Bifidobacterium* spp.	+
	*Fuscobacterium* spp.	+
	*Lactobacillus* spp.	++
	*Peptococcus* spp.	+++
	*Preptostreptococcus* spp.	+++
	*Proprionibacterium* spp.	+
	*Veillonella* spp.	+

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
