# Peer review of "The Vaginal Microbiota, Bacterial Biofilms and Polymeric Drug-Releasing Vaginal Rings"

_pharmaceutics, 2021, doi:10.3390/pharmaceutics13050751_

Round 1

Reviewer 1 Report

This is a rather comprehensive review that describes drug-releasing vaginal rings and vaginal microbiome. I just have several technical suggestions on how to improve the manuscript before publication.

Since the title starts with vaginal microbiota, that should also be an opener for the Abstract. Furthermore, the Introduction section starts too abruptly, without the adequate definition that polymeric ring devices for vaginal use provide controlled delivery of drugs for intravaginal administration over protracted periods of time.

The official rules of microbial nomenclature have to be followed diligently throughout the paper (which is currently not the case in some instances). More specifically, it is pivotal to italicize family, genus, species, and subspecies in all parts of the manuscript (for example, this is not the case in line 252 and lines 270-271). Furthermore, some microorganisms are stated erroneously (Neisseria gonorrhoea instead of Neisseria gonorrhoeae in line 133).

Likewise, when newly proposed Gardnerella species are mentioned in lines 178-179, they should be stated with "sp. nov." designation (i.e., Gardnerella piotii sp. nov., Gardnerella swidsinskii sp. nov. and Gardnerella leopoldii sp. nov.) until formal taxonomic discussions are finished.

The manuscript was submitted with tracked changes, which should be amended.

Reviewer 2 Report

  1. Please supplement scale bars in Fig. 1 to make readers easy to catch the sizes of marketed VRs.
  2. A concise illustration about the market size of VRs would benefit to arouse interests of readers.
  3. As bacterial adherence and biofilm formation would change intravaginal behaviors of VRs, some discussions about the effects on drug release and efficacy are necessary in this review.
  4. From this review, I consider that bacterial adherence on surface of VRs are commonly harmful. If so, what might be the probable solution according to publications up to now?
